# Research Progress in Nonlinear Ultrasonic Testing for Early Damage in Metal Materials

**DOI:** 10.3390/ma16062161

**Published:** 2023-03-08

**Authors:** Xiaoling Yan, Houpu Wang, Xiaozhi Fan

**Affiliations:** 1School of Artificial Intelligence, Beijing Technology and Business University, Beijing 102488, China; 2School of Media and Design, Beijing Technology and Business University, Beijing 102488, China

**Keywords:** nonlinear, ultrasonic wave, early damage, non-destructive testing

## Abstract

There are some limitations when conventional ultrasonic testing methods are used for testing early damage in metal parts. With the continuous development of acoustics and materials science, nonlinear ultrasonic nondestructive testing technology has been used for testing of early damage in metal materials. In order to better understand the basic theory and research progress of the nonlinear ultrasonic testing technology, the classical nonlinear ultrasonic theoretical models, including the dislocation monopole model, dislocation dipole model, precipitate-dislocation pinning model, and contact nonlinear ultrasonic theory-microcrack model, are analyzed in depth. This paper introduces the application and research progress of nonlinear ultrasonic detection technology, which is derived from different acoustic nonlinear effects, such as higher harmonic, wave mixing and modulation, sub-harmonic, resonance frequency spectrum analysis, and non-linear ultrasonic phased array imaging. The key technologies and problems are summarized to provide a reference for the further development and promotion of nonlinear ultrasonic non-destructive testing technology.

## 1. Introduction

Conventional nondestructive testing methods such as ultrasonic testing [1,2], eddy current testing [3,4], X-ray testing [5,6], penetrant testing [7], and magnetic powder testing [8] are effective in detecting macroscopic structural defects in metal materials such as cracks, pores, inclusions, etc. However, in the detection of early damage in metal materials, conventional nondestructive testing methods prove to be dissatisfactory. Macroscopic defects such as cracks appear in metal materials subjected to multiple fatigue alternating loads. The appearance of cracks is the result of multiple damage accumulation in metal materials. In most cases, the duration before crack formation accounts for more than 80% of the total fatigue life of metal materials. In other words, when cracks occur, the remaining fatigue life of parts may be less than 20%. Studies in the 1980s have demonstrated that crack initiation can be defined with a crack size of about 10 micrometers (using the replica method [9]). Then the crack propagation is well predicted by the fracture mechanics. However, the replica method is not suitable to detect and follow cracks in industrial components. Thus, accurate and reliable methods such as nonlinear ultrasonic techniques are needed.

The main reason for the failure of metal parts is the fatigue fracture of metal parts. The microstructure analysis shows that the fatigue damage of the parts can be divided into three stages: In the first stage, many dislocation groups are produced, and the persistent slip bands and microcracks are formed; in the second stage, microcracks nucleate and grow, and macrocracks are produced; in the third stage, the metal parts will eventually undergo fatigue fracture. Ultrasonic nondestructive testing technology can effectively evaluate the macrocracks of metal parts due to fatigue damage, but it is not sensitive to the microcracks generated at the early stage of fatigue damage. When the defect size is less than half of the detected ultrasonic wavelength or the acoustic impedance difference between the defect and the surrounding medium is very small, it is difficult for the traditional ultrasonic nondestructive testing technology to effectively detect it [10,11]. Compared with traditional ultrasonic testing technology, nonlinear ultrasonic testing technology can essentially reflect the impact of small defects in the material on the ultrasonic propagation process. Even if the damage is very small, when the wave propagates in the material, there will be obvious nonlinear phenomena. such as waveform distortion, higher harmonic generation, sideband formation, [12,13,14,15] etc. Non-destructive testing of early mechanical property degradation of materials can be achieved by using nonlinear ultrasonic testing technology [16].

Due to the unique advantages of nonlinear ultrasonic technology in the early damage detection of metal materials, many domestic and foreign scholars have performed a lot of research and achieved a series of results in recent years. However, at present, there is relatively little comprehensive work on this technology. In order to better understand the basic principle of nonlinear ultrasonic testing technology, the research status and progress of this technology in the field of early damage detection of metal materials are introduced. The ultrasonic nonlinear technology is summarized from the aspects of higher harmonic [17,18,19] beam mixing and modulation [20,21,22], sub-harmonic [23], resonance frequency spectrum analysis [24], ultrasonic phased array nonlinear imaging [25] in this paper. The key technologies and problems are summarized. The development trend of this technology is anticipated in order to provide a valuable reference for further research and application of nonlinear ultrasonic testing technology.

## 2. Basic Theory of Nonlinear Ultrasonics

### 2.1. Generation of Higher Harmonic

The nonlinear wave equation in solid materials was established by Cantrell [26]. According to the nonlinear wave equation, the essential reason for the nonlinear acoustic effect caused by the damage (micro-cracks, pores, and lattice defects) in material can be described.

The wave equation for the one-dimensional longitudinal wave is as follows:(1)$$\rho \frac{\partial^{2}u}{\partial t^{2}}=\frac{\partial \sigma }{\partial x}$$
where, *ρ* is the density, *u* is the displacement, *σ* is the normal stress. When the deformation of the material is small enough, the normal strain *ε* is expressed as:(2)$$\varepsilon =\frac{\partial u}{\partial x}$$

According to nonlinear Hooke’s law, the stress–strain relation can be expressed as:(3)$$\sigma =E\varepsilon \left(1+\frac{1}{2}\beta \varepsilon +\frac{1}{3}\delta \varepsilon^{2}\ldots \right)$$
where, *E* is young modulus, *β* and δ are the second- and third-order nonlinear coefficients, respectively. According to Equations (1) to (3), the nonlinear wave equation can be deduced (the high-order terms in the formula are ignored) as:(4)$$\rho \frac{\partial^{2}u}{\partial t^{2}}=E\frac{\partial^{2}u}{\partial x^{2}}+E\beta \frac{\partial u}{\partial x}\frac{\partial^{2}u}{\partial x^{2}}$$
where, *t* is the time (s), and *x* is the distance of wave propagation (m). Set the preliminary condition as follows:(5)$$u\left(0,t\right)=A_{1}\sin\left(\omega t\right)$$

The perturbation method is applied to deduce the approximate solution of nonlinear wave Equation (4):(6)$$u\left(x,t\right)=A_{1}\sin\left(kx-\omega t\right)+\frac{1}{8}\left(A_{1}{}^{2}k^{2}\beta x\right)\cdot \cos2\left(kx-\omega t\right)$$

Where, *A*_1_ is the amplitude of fundamental wave, *ω* is the frequency, and *k = w/c* is the wave number, *A*_2_ is the amplitude of second harmonics, and *A*_2_ is expressed as:(7)$$A_{2}=\frac{1}{8}\left(A_{1}{}^{2}k^{2}\beta x\right)$$

According to Equation (7), the expression of *β* is:(8)$$\beta =8\left(\frac{A_{2}}{A_{1}{}^{2}}\right)\frac{1}{k^{2}x}$$

According to the aim of ultrasonic testing for the damage in the measured part, the *x* and *k* are usually constants. Thus, *β* is simplified as:(9)$$\beta =\frac{A_{2}}{A_{1}{}^{2}}$$

### 2.2. Classical Nonlinear Ultrasonic Theoretical Models

The ultrasonic nonlinear coefficient is not only related to the crystal structure of metal materials but also to the local strain of materials, that is, the microstructure of materials, such as dislocations, cracks, and precipitates [27,28,29,30], has greater influence on the acoustic nonlinear coefficient than the lattice strain of materials. This section summarizes the classical nonlinear ultrasonic theoretical models and introduces different theoretical analysis models, including the dislocation monopole model, the dislocation dipole model, and the precipitation-pinned dislocation model.

#### 2.2.1. Dislocation Monopole Model

Dislocations refer to the internal microscopic defects of crystal materials, that is, the local irregular arrangements of atoms (crystallographic defects). Multiple dislocations will interact with each other under external load and form many pinning points. Suzuki et al. [31] first proposed the dislocation model that produces the nonlinear effect mechanism, and then Hikata et al. [32] further expanded it. This model can be used to explain the mechanism of second harmonic generation in ultrasonic testing. Figure 1 shows the dislocation monopole model. The dislocation chord is pinned between M and N points, and the distance between the two points is 2L. The pinning points can be other dislocations or point defects in materials [33]. Tangential stress *τ = R σ*, *R* is the conversion coefficient of shear stress and normal stress, *σ* is normal stress, *τ* acting on the staggered chord MN, the convex bow chord MIN is formed between the pinning points, and *b* is the Burgers vector, stress *τ* is not enough to separate the dislocation string from the pinning points M and N. The shaded area in Figure 1 is the area swept by the dislocation when it moves in the slip plane. Figure 2 shows the dislocation movement under stress.

According to the Hikata model, the total strain of the material is the sum of lattice strain and strain caused by dislocation movement:(10)$$\varepsilon =(\frac{1}{E_{1}}+\frac{2}{3}\frac{\Omega \Lambda L^{2}R}{\mu })\sigma +\frac{E_{2}}{E_{1}^{3}}\sigma^{2}+\frac{4}{5}\frac{\Omega \Lambda L^{4}R^{3}}{\mu^{3}b^{2}}\sigma^{3}+\ldots$$
where *E*_1_ and *E*_2_ are the second- and third-order elastic constants, *Λ* is the dislocation density, *L* is the dislocation line length, and *R* is the conversion coefficient of shear stress and normal stress, *μ* is the shear modulus, *σ* is the applied stress, *b* is the Burgers vector, and *Ω* is the conversion coefficient between tangential strain and normal strain. When minor changes occur to stress *σ*, the stress change corresponding to the strain caused by dislocation movement is:(11)$$\Delta \sigma =\frac{\partial \sigma }{\partial \varepsilon }\Delta \varepsilon +\frac{1}{2}\frac{\partial^{2}\sigma }{\partial \varepsilon^{2}}\Delta \varepsilon^{2}+\frac{1}{6}\frac{\partial^{3}\sigma }{\partial \varepsilon^{3}}\Delta \varepsilon^{3}+\ldots ={\left[\frac{1}{E_{1}}+\frac{2}{3}\frac{\Omega \Lambda L^{2}R}{\mu }\right]}^{-1}\Delta \varepsilon -(\frac{E_{2}}{E_{1}^{3}}+\frac{12}{5}\frac{\Omega \Lambda L^{4}R^{3}}{\mu^{3}b^{2}}\sigma )/{\left(\frac{1}{E_{1}}+\frac{2}{3}\frac{\Omega \Lambda L^{2}R}{\mu }\right)}^{3}\Delta \varepsilon^{2}+\ldots$$

The nonlinear stress-strain relationship is:(12)$$\sigma =A\varepsilon +\frac{B}{2}\varepsilon^{2}$$

It can be seen from Formulas (11) and (12)
(13)$$A={\left[\frac{1}{E_{1}}+\frac{2}{3}\frac{\Omega \Lambda L^{2}R}{\mu }\right]}^{-1}$$
(14)$$B=-[2(\frac{E_{2}}{E_{1}^{3}}+\frac{12}{5}\frac{\Omega \Lambda L^{4}R^{3}}{\mu^{3}b^{2}}\sigma )/{\left(\frac{1}{E_{1}}+\frac{2}{3}\frac{\Omega \Lambda L^{2}R}{\mu }\right)}^{3}]$$

Substitute Formulas (11) and (12) into the second harmonic amplitude formula, *A*_2_=1/8(*B/A*)(*A*_1_*k*)^2^*x*, and we obtain:(15)$$A_{2}=\frac{1}{4}[(\frac{E_{2}}{E_{1}^{3}}+\frac{12}{5}\frac{\Omega \Lambda L^{4}R^{3}}{\mu^{3}b^{2}}\sigma )/{\left(\frac{1}{E_{1}}+\frac{2}{3}\frac{\Omega \Lambda L^{2}R}{\mu }\right)}^{2}]{\left(A_{1}k\right)}^{2}x$$

Considering fundamental wave attenuation *α*_1_ and second harmonic attenuation *α*_2_, 1/*E*_1_ is far greater than 2*ΩΛL^2^R/*(3*μ*), So Formula (15) can be simplified as follows
(16)$$A_{2}=\frac{1}{4}[(\frac{E_{2}}{E_{1}}+\frac{12}{5}\frac{E_{1}^{2}\Omega \Lambda L^{4}R^{3}}{\mu^{3}b^{2}}\sigma ){\left(A_{1}k\right)}^{2}\frac{e^{-2}{}^{\alpha_{1}x}-e^{-}{}^{\alpha_{2}x}}{\alpha_{2}-\alpha_{1}}$$

According to the above theoretical derivation, the dislocation in the material microstructure will cause the generation of the second harmonic in the ultrasonic testing process. The second harmonic amplitude is proportional to stress, dislocation density, and the fourth power of dislocation line length. *E*_2_*/E*_1_ is the effect of lattice strain on the second harmonic amplitude, *12E*_1_*^2^ΩΛL^4^R^3^/*(*5μ^3^b^2^*) is the effect of dislocation on the second harmonic amplitude.

#### 2.2.2. Dislocation Dipole Model

With the deepening of material fatigue, the complex dislocation structures of dislocation dipole and multipole will appear. A dislocation dipole is a common micro-defect in crystal materials. When the distance between the slip planes of two different edge dislocations with parallel slip planes is 300~400 Å, under the action of attraction, one dislocation moves to the top of the other dislocation to achieve a certain equilibrium and forms a dislocation dipole. As shown in Figure 3 [34], on the xoy plane, medium I with a shear modulus of *G*_1_ and medium II with a shear modulus of *G*_2_ occupy the upper half plane (S+) and lower half plane (S−), respectively. The center of the screw dislocation dipole is in the medium *z*_0_ (*z*_0_ = *ρe^iθ^*), and it contains two screw dislocations located at *z*_1_ (*z*_1_ *= z*_0_ *− ae^iφ^*) and *z*_2_ (*z*_2_ *= z*_0_
*+ ae^iφ^*) points, whose Burger’s vectors are *b*_1_ (*b*_1_ *= bz*) and *b*_2_ (*b*_2_*= −bz*) respectively. The dipole distance is 2a. Cantrell’s research [34,35] shows that when the dislocation density is large, the ultrasonic second harmonic amplitude is not only related to the dislocation line length and stress but also related to the dislocation arrangement in the material microstructure. The effect of the dislocation dipole on the nonlinear effect can be expressed [33,34] as
(17)$$\beta^{dp}=\frac{16\pi^{2}\Omega \Lambda_{dp}R^{2}h^{3}{\left(1-\nu \right)}^{2}E_{1}^{2}}{\mu^{2}b}+\frac{384\pi^{3}\Omega \Lambda_{dp}R^{3}h^{4}{\left(1-\nu \right)}^{3}E_{1}^{2}}{\mu^{3}b^{2}}\sigma$$
where *Λ_dp_* is the density of the dislocation dipole, *h* is the distance of the dislocation dipole, and *ν* is Poisson’s ratio.

The total nonlinear coefficient caused by lattice strain, dislocation monopole, and dislocation dipole displacement [26,33,34,35,36,37,38] is:(18)$$\beta =\frac{\beta^{e}+f^{mp}\beta^{mp}+f^{dp}\beta^{dp}}{{\left(1+f^{mp}\Gamma^{mp}+f^{dp}\Gamma^{dp}\right)}^{2}}$$
where *β^e^* is the nonlinear coefficient of lattice strain, *β^mp^* is the nonlinear coefficient of dislocation monopole, *β^dp^* is the nonlinear coefficient of dislocation dipole, *f^mp^* and *f^dp^* are the volume percentages of the dislocation monopole and the dislocation dipole contained in the material, *Γ^mp^* and *Γ^dp^* are constants related to material properties.

#### 2.2.3. Precipitate-Dislocation Pinning Model

The precipitate in the microstructure of the material has no significant effect on the nonlinear coefficient, but the precipitate embedded in the matrix will produce localized stress around the matrix. If this stress is applied to the pinned dislocation line, it will have a great impact on the nonlinear coefficient. The precipitates with a radius of *r_p_* are embedded in the matrix. The natural radius of the matrix is *r_a_*, and the relationship *r_p_ = r_a_* (1 *+ δ*), *δ* is the mismatch parameter between the matrix lattice and precipitate. As shown in Figure 4, the precipitates of the sphere are embedded in the isotropic matrix medium, and the resulting local pressure *P* is applied to the lattice. When *r >r_a_*, the stress around the lattice is *σ_r_*. *σ_r_* can be expressed as [38]
*σ_r_ = −p r_p_^3^/r^3^* (19)

The elastic properties of precipitate and matrix are different, and the stress produced by precipitate at matrix radius *r* is:(20)$$\sigma_{r}=-4\mu \delta \frac{3B_{P}}{3B_{P}+4\mu }\frac{r_{p}^{3}}{r^{3}}$$
where *B_P_* is the bulk modulus and *µ* is the shear modulus. As shown in Figure 4, the influence of the precipitation-dislocation pinning model on the nonlinear coefficient can be expressed as follows [38]: (21)$$\Delta \beta_{ppd}\approx 307\frac{3B_{P}}{3B_{P}+4\mu }\frac{\Omega R^{3}E_{1}{}^{2}}{\mu^{2}b^{2}}(\Lambda \frac{r_{p}^{3}}{N^{1/3}}\left|\delta \right|)$$
where *N* is the number density of precipitates.

### 2.3. Contact Nonlinear Ultrasonic Theory Model

Generally, the increase in dislocation density in the later stage of fatigue of metal components will lead to the initiation of microcracks. Figure 5 is the schematic diagram of the nonlinear ultrasonic response of microcracks. The crack surfaces move under the action of ultrasonic waves and result in a dynamic force *G*(*y*). The crack spacing *y*(*t*) changes under the action of ultrasonic waves, and the crack interfaces will also move in the range of *y*(*t*). Generally, *y*(*t*) is uneven. When an ultrasonic wave passes through this partially closed crack, distortion will occur, resulting in the generation of the second harmonic, and the tension/compression effect produced by the two interfaces of elastic rough contact is the main source of nonlinearity. With the propagation of the microcrack, the nonlinear acoustic effect is more obvious. Nazarov deduced the theoretical model of the nonlinear coefficient of microcracks in metal materials. Cantrell used this model to determine the influence of fatigue cracks on nonlinear effects. The results show that the value is related to fatigue process parameters and fatigue degree. Relevant literature [26,30,39,40,41] shows that the nonlinear effect caused by microcracks is larger than that caused by dislocations.

Where A and B represent the rough interface of the micro crack, *P* represents the static closing force, *c* is the velocity of the ultrasonic wave, *t* is the time, *x* is the displacement, *y*(*t*) is the crack spacing, and *G*(*y*) is the dynamic force of the joint surface generated by the mutual movement of A and B. *u*(−0, *t*) is the equivalent interface of the rough interface A, *u*(+0, *t*) is the equivalent interface of the rough interface B. The incident wave *f*(*x* − *ct*) incidents on the rough interface A. *g*(*x* + *ct*) is the reflected wave, and *h* (*x* − *ct*) is the transmitted wave.

In conclusion, nonlinear ultrasonic theory can be divided into two categories: classical nonlinear and contact nonlinear. The classical nonlinear ultrasonic theory starts with the discrete model and establishes the nonlinear ultrasonic wave equation in solid media. By solving the wave equation, the relationship between the nonlinear coefficient and the amplitude of the fundamental wave and the second harmonic is obtained. In addition, the classical nonlinear ultrasonic theory clarifies the influence mechanism of dislocation monopoles, dislocation dipoles, and precipitates in the material microstructure on the ultrasonic nonlinear effect and establishes the mathematical model of the ultrasonic nonlinear coefficient for different types of microscopic defects. The research in contact nonlinear theory mainly focuses on the ultrasonic nonlinear response on the solid interface, and the change of the spacing between two rough contact interfaces is studied, which is related to the dynamic force of the interface and the acting force of ultrasonic. The fundamental reason for the nonlinear harmonic response is demonstrated. The development and perfection of nonlinear ultrasonic theory have laid a solid foundation for the engineering application of this technology.

## 3. Application of Nonlinear Ultrasonic Testing Technology

### 3.1. Higher Harmonic Technology

High-order harmonic detection technology has been widely used in closed crack [41,42], interface debonding, composite bonding quality [43], weld quality [44], material mechanical property degradation [45], crystal structure symmetry [46], environmental corrosion, early fatigue damage [47], and life prediction. The detection principle of higher harmonic technology is shown in Figure 6. When a single-frequency excitation signal with the same amplitude is used to detect the defect-free and defective specimens, the detection signal of the defect-free specimen is still a single-frequency wave, but the amplitude is reduced, and the detection signal of the specimen with micro-defects is distorted. After transforming the distorted signal into the frequency domain, it is found that there are higher harmonic components in the frequency domain besides the fundamental wave. These higher-harmonic signals contain information about defects in the specimen, and the purpose of characterizing the defects in the specimen can be achieved by using higher-harmonic technology.

Because the ultrasonic energy is concentrated near the surface of the sample, the surface wave is more sensitive to the early damage and thermodynamic defects of the medium surface, and the nonlinear effect is obvious. As shown in Figure 7, the nonlinear Rayleigh wave measurement setup contains one narrow band piezoelectric transducer that excites a longitudinal wave, which is introduced in an acrylic wedge to launch a Rayleigh surface wave in the specimen. Thiele et al. [48] detected the precipitation-hardening material 17-4 PH stainless steel using Rayleigh waves, 2.1 MHz is used for the excitation of the transducer, and the receiving transducer was centered at 3.9 MHz. The obtained output signal is amplified by 40 dB to improve the signal-to-noise ratio. The material is thermally treated to obtain different precipitation stages, and then the influence of the precipitates on the acoustic nonlinearity parameter is assessed. The results show that the nonlinear parameter of the Rayleigh waves decreases with the heat treatment time of the solution-annealed and air-cooled specimens. The drop from the unaged specimen SA-AC to specimen SA-AC 400-1.0 (which is aged at 4000C for 1 h) and specimen SA-AC 400-6.0 (which is aged at 4000C for 6 h) is significant, exceeding 30% and 40%, respectively, and it can therefore be concluded that nonlinear ultrasonic is sensitive to the microstructural changes associated with precipitation. Li et al. [49] used this technology to detect the tensile fatigue damage and corrosion fatigue damage of Q235 steel, with the excitation and receiving transducers centered at 5 MHz. *β*/*β*_0_ (*β*_0_ is the ultrasonic nonlinear coefficient of the specimen not subjected to fatigue loading, *β* is the ultrasonic nonlinear coefficient of the specimen subjected to fatigue loading) is used to regularize the ultrasonic relative nonlinear coefficient. The test results are shown in Figure 8. The loading frequencies of specimen 1, specimen 2, and specimen 3 are 15 Hz, 20 Hz, and 20 Hz, respectively. The specimen 3 has been soaked in 10% salt water for 20 days, and specimen 1 and specimen 2 are not subject to corrosion treatment. The change trend of the relative nonlinear coefficient of the No. 1 and No. 2 specimens is basically the same, and they remained unchanged after 400,000 fatigue cycles, which indicates that fatigue loading frequencies have little effect on the fatigue damage degree of the test specimen. The relative nonlinear coefficient of the No. 3 specimen is generally larger than that of the No. 1 and No. 2 specimens. It remained unchanged after 300,000 fatigue cycles, earlier than the No. 1 and No. 2 test pieces, indicating that the corrosion environment aggravated the fatigue damage degree of the test pieces.

When higher harmonic technology is applied, ultrasonic incidence methods include the oblique incidence method, end-face incidence method, and direct incidence method. Relevant research [47,48,49] show that the end-face incident method has a long detection distance, large signal attenuation, and low detection accuracy; the direct incidence method has a short detection distance and a small detection range; the oblique incidence method can not only control the detection distance but also ensure that the whole fatigue area can be detected, and the precision of detecting micro-defects is high. Wan [43] carried out fatigue loading tests on U71Mn steel specimens and extracted the nonlinear coefficients in the specimens using the above-mentioned three incident methods. The results show that the oblique incidence method can not only ensure the interaction between the acoustic wave and the fatigue damage zone but also have good adaptability to different welded joints and can obtain better nonlinear ultrasonic testing results.

Higher harmonic technology also has the potential to predict failure at the early stage of fatigue when the dislocation starts to accumulate. Zhang et al. [50] carried out intermittent low-cycle fatigue tests on 304 austenitic stain-less steel using transmission electron microscopy and higher harmonic technology. The research results show that the increase in ultrasonic nonlinear coefficient in the first stage of low cycle fatigue of 304 stainless steel is mainly due to the increase in dislocation monopole and dipole density, while the decrease in ultrasonic nonlinear coefficient in the second stage of fatigue is mainly due to the decrease in dislocation monopole density and dipole height during the formation of dislocation cells. Therefore, the ultrasonic nonlinear response of the complex dislocation structure in the fatigue process needs to be further explored. Qiao et al. [51] carried out intermittent fatigue loading tests on 316L stainless steel specimens formed by Selective Laser Melting (SLM) and evaluated their fatigue damage using higher harmonic technology. The nonlinear ultrasonic testing system used in the test is shown in Figure 9. The frequencies of the transmitting signal and the receiving signal are set to 5 MHz and 10 MHz, respectively. The tension-tension fatigue tests are performed on the specimen; when the fatigue cycle number is an integral multiple of 5000, the fatigue loading testing system is suspended; meanwhile, the tensile load remains constant, and the nonlinear ultrasonic tests for the specimen are executed. Figure 10 shows the “mountain curve” between the normalized nonlinear coefficient (*β/β*_0_) and the fatigue cycles. Firstly, *β/β*_0_ increases within 0~38,000 fatigue cycles (about 0~56% of the fatigue life fraction). The multiplication of dislocations and the emergence and propagation of micro-cracks are the reasons for the increase in the normalized ultrasonic nonlinear coefficients. Then *β/β*_0_ gradually decreased within 38,000~68,000 fatigue cycles (about 56~100% of the fatigue life fraction), because the microcracks had expanded into macrocracks at this stage. The nonlinear ultrasonic testing results are in good agreement with the nonlinear ultrasonic theoretical model and the Transmission Electron Microscope (TEM) results (as shown in Figure 11) and scanning electron microscope (SEM) experimental analysis results. (as shown in Figure 12).

In conclusion, higher harmonic technology has played a key role in the early damage detection of metal materials. However, in the process of higher harmonic detection, the nonlinear coefficient represents the average value of the area between the transmitting transducer and the receiving transducer [52,53,54], and the spatial resolution of detection is greatly limited. In addition, transducers, coupling agents, and experimental equipment will also produce nonlinearity, which increases the difficulty of distinguishing experimental data. Therefore, signal processing and analysis [55] are the key technologies to ensure the reliability of nonlinear ultrasonic testing results. It is necessary to strengthen the theoretical research on signal processing. In the face of such technical defects, modulation detection technology came into being.

### 3.2. Modulation Technology

The basic principle of the modulation technology is based on the interface nonlinear response model [56] shown in Figure 13. Metal material defects are usually rough elastic surface contacts, and the effect of force will cause local damage to the contact surface, thus generating nonlinear elasticity on the interface (the intact interface is approximately linear elasticity). When the sound wave with a certain frequency acts on the defect area, the contact interface will jump, which is a nonlinear response different from the response of the non-defect interface area. At this time, another incident wave can be used to detect this nonlinearity. The interface nonlinear response model shows that the defect is the source of the nonlinear wave, which can convert part of the applied sound energy into nonlinear sound waves with different frequencies. Therefore, the basic principle of the modulation technology is to make the nonlinear response of defects more obvious by adding the modulation wave, and then use the scanning wave to capture this nonlinearity.

Figure 14 shows the detection principle of the vibration modulation technology, which uses only one ultrasonic wave as the scanning wave. This method is different from the frequency-mixing method. The vibration modulation method can be used to detect complex structural parts. Because the applied vibration or impact frequency is low and acts on the whole sample, there is no special requirement for the installation position of the transmitting and receiving transducers. This method has the potential to detect large components and the remote end of the structure. If there are defects in the test specimen, the spectrum of the received signal will have new frequency components. According to the existence of harmonics and the magnitude of their amplitude, the existence and size of defects can be determined.

The advantage of the modulation technology is that the nonlinear response of the defect is strengthened by applying a low-frequency modulation wave to the whole specimen to be tested. Therefore, it is possible to distinguish the nonlinear signal of the material layer, boundary, transducer, coupling agent, and other nonlinear signals from the nonlinear signal of the defect. In addition, because the low-frequency sound wave is applied to the whole specimen, the installation positions of the transmitting and receiving transducers are not sensitive during the detection. The defects of any shape and in any position can be detected. Low-frequency wave propagation is stable, and attenuation is small, which can be used to detect large structural parts and remote parts of the structure. The modulation method has been applied to the bonding quality detection of thermoplastic plates and composite structures [57,58,59].

### 3.3. Frequency-MixingTechnology

Since the 1960s, many scholars [60,61,62,63,64,65,66] have found that when two incident waves propagate in nonlinear media, if certain conditions are met, the two waves will interact and produce a new wave. The frequency of the new wave is different from that of the two incident waves, that is, the frequency-mixing wave. When there are no defects in the detected area, the two incident waves meet the linear superposition principle, that is, the third wave with a new frequency will not be generated. If there are microscopic defects, such as closed cracks, in the detected region, the two incident waves will interact and generate a new wave, and the frequency of the new wave is equal to the sum or difference of the frequencies of the two incident waves.

Figure 15 shows the diagram of the interaction between two waves in a nonlinear medium. Table 1 [67] shows the mixing conditions of the two waves. *ψ* represents the included angle of two incident waves *k*_1_ and *k*_2_ (*k*_1,_ *k*_2_ are the wave numbers of the incident wave), and *γ* is the included angle between the incident wave *k*_1_ and the mixing wave *k*_3_. In Table 1, *L* and *T* represent longitudinal and transverse waves, respectively, α is the frequency ratio *ω_l_/ω*_2_ (*ω*_1_*, ω*_2_ are respectively the frequencies of the two incident waves *k*_1_ and *k*_2_); and *c* is the velocity ratio *c*_t_/*c_l_* (*c_t_* is the velocity of the shear wave, *c_l_* is the velocity of the longitudinal wave). When the two waves meet in this area, they will interact and produce the third wave. The frequency of the third wave is equal to the sum or difference of the frequencies of the two incident waves. Analyzing this frequency component can effectively detect and evaluate the fatigue damage of materials.

According to the data in Table l, the frequency ratio α and the included angle of the fundamental wave *ψ* can directly affect the propagation direction of the frequency-mixing wave *γ*. By adjusting the frequency ratio, the propagation direction of the frequency-mixing wave can be controlled. Demchenko et al. [68] simulated the change of the scattering coefficient, the interaction angle of two waves, and the scattering angle of the frequency-mixing wave with the frequency ratio in PVC materials. The simulation results provide a reference for selecting the appropriate frequency and incident angle of the incident wave.

#### 3.3.1. Collinear Wave Mixing Technology

When collinear frequency-mixing technology is applied, two incident waves are either parallel to each other or on the same axis. According to the type of incident wave, collinear wave mixing technology [69,70] can be divided into collinear longitudinal wave mixing technology, collinear shear wave and longitudinal wave mixing technology, and Lamb wave mixing technology. Theoretical research [71] shows that the propagation range of the mixing wave generated by the interaction of two collinear incident shear waves in the nonlinear region is limited to the mixing region and that it cannot propagate freely outside the mixing region, so the mixing of two collinear incident shear waves is not considered.

The principle of collinear wave mixing detection is shown in Figure 16. Two transducers are installed at both ends of the specimen to ensure that the transmission direction of the excitation signal emitted by the two transducers is collinear or parallel and the transmission frequency is, respectively, *ω*_1_*,* and *ω*_2_. When two excitation signals meet in the test specimen, if there is slight damage in the test specimen, the two excitation signals will interact with the damaged part and generate a new signal (frequency: *ω*_1_ *± ω*_2_). This signal reflects the degree of fatigue damage at the location where the two excitation signals meet.

Zhang et al. [50] used collinear longitudinal wave mixing technology to detect and explore the fatigue damage of 40Cr steel. The results show that this method can effectively evaluate the fatigue damage of the specimen. In addition, this method can effectively avoid the nonlinearity caused by the test instrument because the nonlinearity caused by the test instrument is mainly manifested as the second harmonic of the excitation signal, while the collinear wave mixing detection uses the sum or difference frequency of the two excitation signals. Tang et al. [72] used collinear shear wave and longitudinal wave mixing technology to detect the local plastic deformation damage of Al6061. By controlling the transmission time of the fundamental frequency pulse wave, the two waves meet at the specified position, and scanning of the test specimen can be realized. The experimental results show that the resonance amplitude of the specimen with plastic deformation is greater than that of the intact specimen at the corresponding position; the more obvious the plastic deformation of the specimen is, the greater the resonance amplitude is. Thus, without changing the position of the transducers, the local area of the specimen can be detected to achieve the purpose of material damage location and greatly simplify the detection process. However, this technology has a high requirement for the size of the mixed area. The size of the mixed area should be equal to the size of the test area of the specimen. A too large mixing area will affect the spatial resolution of scanning and reduce the accuracy of damage location, while a too small mixing area will affect the signal extraction. Jiao et al. [73] used nonlinear Lamb wave mixing technology to detect cracks in cold-rolled carbon steel plates. The results show that the nonlinear coefficient is highly sensitive to cracks in the test specimen.

Compared with higher harmonic technology, collinear wave mixing technology has unique advantages. Firstly, collinear wave mixing technology reduces the influence of the nonlinearity of the experimental instrument itself on the measurement results. Secondly, without changing the position of transducers, the purpose of comprehensive scanning and damage location of the specimen can be achieved by adjusting the emission time of the two incident waves, which simplifies the detection process. However, the collinear wave mixing technology also has certain limitations. Because the technology requires that the two incident waves be on the same axis, it is difficult to arrange the transducers for the larger specimen in the actual operation process. In addition, the shear wave and longitudinal wave mixing technology requires high frequency conditions with the two incident waves, and the intensity of the resonance wave generated is weak, which is easy to be affected by noise. Therefore, in order to receive the resonance wave signal more accurately and effectively, the nonlinear ultrasonic detection system should have stronger power amplification and weak harmonic signal detection capabilities.

#### 3.3.2. Non-Collinear Wave Mixing Technology

The principle of non-collinear wave mixing detection is that two non-parallel waves are incident on the test specimen, and when the two incident waves meet the acoustic resonance conditions, the third wave will be generated [73,74]. By adjusting the incident angle and relative position of the wave, the meeting position of two waves in the test specimen can be controlled, and the scanning of damage at different positions can be realized. Therefore, compared with collinear wave mixing technology, non-collinear wave mixing technology can select the detection area more flexibly. The non-collinear wave mixing technology can be divided into three categories according to the type of incident wave: non-collinear longitudinal wave mixing technology [75], non-collinear shear wave mixing technology [76,77], and non-collinear longitudinal wave and shear wave mixing technology [78].

Figure 17 is the schematic diagram of the test device for non-collinear wave mixing detection. The Ritec-SNAP system outputs two longitudinal wave signals with different frequencies, which are incident on the wedge at the specified angle and excite the transverse waves. Then the transverse waves incident on the test piece at a certain angle interact with each other to generate the frequency-mixing wave within the specified scanning range. The receiving transducer at the other end of the test specimen receives the frequency-mixing signal, which is amplified by the preamplifier and then received by receiving channel 1 of the SNAP system.

Thanseer et al. [79] used the non-collinear shear wave mixing technology to detect the closed crack of a 9Cr-Mo steel specimen. By adjusting the relative position of the transmitting transducers, the detection of the specimen along the vertical and horizontal directions can be realized. In terms of plastic deformation detection of materials, Croxford et al. [80] carried out ultrasonic detection of Al2014-T4 under fatigue damage and plastic deformation, respectively (non-collinear shear wave mixing technology, two longitudinal wave transducers with a frequency of 5.5 MHz were selected). According to Snell’s theorem, the plexiglass wedge with an angle of 60° is selected to excite the transverse wave. The two transverse waves meet and interact in the test specimen to generate a new longitudinal wave. The new longitudinal wave is received by the receiving transducer. The results indicate that non-collinear wave mixing technology is potentially more attractive for assessing material state than other nonlinear ultrasonic techniques because system nonlinearities can be both independently measured and largely eliminated. Li et al. [81] conducted research on the detection and location of surface damage in 6061 aluminum alloy by non-collinear mixing of one longitudinal wave and one transverse wave at different frequencies. Results show that fatigue damage and surface corrosion can be identified and located by checking the variation of nonlinear frequency-mixing responses. Mora et al. [82] showed theoretically and experimentally how to generate a zero-group velocity (ZGV) guided wave through the sum-frequency interaction of two high-amplitude main waves. ZGV guided wave has proved to be a very sensitive tool for characterizing materials or thickness variations with sub-percent accuracy at space resolutions of about the plate thickness. Zhang et al. [83] inspected the diffusion welding interface of TC4 titanium alloy and realized the evaluation of the quality of the welding interface by analyzing the non-collinear frequency-mixing signal reflected by the interface. Blanloeuil et al. [84] conducted simulation research on aluminum specimens with closed cracks based on the finite element method. The effects of the incidence angle of shear waves and pressure on the amplitude and waveform of the frequency-mixing longitudinal wave were analyzed. The finite element model demonstrates that non-collinear shear wave mixing technology is effective and promising when applied to a closed crack. The scattering of the frequency-mixing longitudinal wave also makes it possible to image the crack, giving its position and size.

Compared with the higher harmonic and collinear frequency-mixing technologies, the non-collinear frequency-mixing technology [80,81,82,83,84,85] reduces the nonlinear interference generated by the instrument; In addition, the position of the transmitting transducer can be adjusted to scan the test specimen and locate the defect, and the propagation direction of the frequency-mixing wave can be controlled by adjusting the frequency and angle of the incident wave. However, non-collinear wave mixing technology also has certain limitations. Firstly, because two transmitting transducers and one receiving transducer need to be arranged in the actual operation process, the size and shape of the specimen should meet the layout requirements of the transducer; Secondly, in the process of scanning the specimen, some edges of the specimen will be in the blind area and cannot be detected; Finally, the received signal quality is poor, and the signal-to-noise ratio needs to be improved.

### 3.4. Sub-Harmonic Technology

The traditional nonlinear effect refers to the higher harmonic generated by the wave propagating in the medium that meets nonlinear Hooke’s law. The sub-harmonic and DC effects can be called the non-traditional nonlinear effects. As shown in Figure 18, the sub-harmonic [86] in the medium is near the wave propagation area, and the nonlinear waveform is distorted. Its typical feature is that the period of the time-domain signal increases to twice that of the original signal and the frequency decreases by half; As shown in Figure 18, the DC effect, also known as the mechanical diode effect, is a nonlinear rectification phenomenon such as amplitude modulation. The nonlinearity mainly comes from the stress-strain nonlinearity of cracks, interfaces, and contact surfaces, which reflects the local defect characteristics of materials. The existence of defects will lead to strong nonlinear distortion when ultrasonic waves interact with them. Because subharmonic and DC effects contain low-frequency band components, sometimes a unified model can be used to describe this phenomenon. Relevant research [87] shows that the amplitude of the subharmonic is proportional to the square root of the DC effect amplitude.

Generally, an oblique incident longitudinal wave is used to act on a closed crack, which will produce subharmonics. The “inertia” effect can explain the mechanism of subharmonics. When the ultrasonic is incident on one interface of a closed crack, the other interface will produce forced vibration due to the interaction between the ultrasonic and the incident interface of the crack, but in fact, the crack is not completely closed at this time, which causes the amplitude of the incident interface and the following interface to be unequal, The period of the receiving signal is increased to twice that of the incident signal, that is, the following interface moves by “inertia” when the crack opens. The DC effect can also be explained by the “inertia” model.

Subharmonics are very suitable for detecting closed cracks. Because the root of subharmonic generation lies in the forced vibration of the two interfaces of the closed crack, the nonlinear effect of coupling agent and instrument is avoided.

### 3.5. Ultrasonic Resonance Spectroscopy Technology

The ultrasonic resonance spectroscopy technology [88] measures the resonant frequency spectrum of the specimen and calculates various elastic constants of the specimen according to its geometric parameters and density. This method is very accurate when the specimen has a regular geometric shape and is a uniform medium. The researchers [89,90,91] found that when the characteristic parameters (e.g., resonance amplitude and resonance peak frequency) of the ultrasonic excitation signal vary, the resonance frequency spectrum of the specimen containing defects presents nonlinear variation, which is not obvious for the specimen without defects. Therefore, the resonance frequency spectrum of the specimen with early damage will change nonlinearly, so the damage of the specimen can be detected using ultrasonic resonance spectroscopy technology.

When ultrasonic resonance spectroscopy technology is applied in detecting and evaluating composite materials [89,90,91], F-scan (Feature-scanning) is used in defect ultrasonic imaging. This technology makes full use of the useful information about defects and materials provided by ultrasonic echo, and the signal is extracted and reconstructed by taking the waveform characteristics such as the rise time, fall time, pulse period, and spectrum characteristics of the waveform or the defect characteristics such as the type, shape, and size of the defect as the feature quantity and finally displayed in the form of imaging. F-scan has the functions of data acquisition, data processing, and automatic evaluation, making it an important technology in modern quantitative nondestructive testing. Rus et al. [91] inspected the defects in CFRP (carbon fiber reinforced polymer) plates using ultrasonic resonance spectroscopy technology. Figure 19 shows the F-scan imaging results of defects in the CFRP plate. The frequency peak value of the thickness resonance of the specimen plate is coded in grayscale for all the scanning positions. Figure 19a exhibits a significant decrease in frequency at scanning positions y higher than 23 mm, where aluminum inclusions are located (num 1). The delamination can be detected as the surface with a higher noise level in Figure 19 (num 2) because almost no ultrasonic of the thickness-resonance frequency range is transmitted through there. A flat-bottom hole is located at num 3 (Figure 19), due to the high intensity of ultrasonic in the thickness-resonance frequency range that are transmitted through the area (Figure 17 num 3). The results demonstrate that ultrasonic resonance spectroscopy technology shows an advantage in characterizing the localized features of the specimens via contact-free ultrasonic inspection.

Above all, ultrasonic resonance spectroscopy has some advantages. Firstly, different types of defects can be detected according to the resonance behaviors of the specimen and the propagation of an ultrasonic wave observed in a single measurement process. Secondly, compared with radiography, it is more sensitive to some features (e.g., delaminations, pores, and closed cracks) that have strong impacts on the mechanical properties of the test specimen. Thirdly, the features of the specimen can be characterized via contact-free ultrasonic inspection. The disadvantages of ultrasonic resonance spectroscopy are that higher data volumes (scan data matrices) need to be processed and used.

### 3.6. Ultrasonic Phased-Array Nonlinear Imaging Detection Technology

Traditional ultrasonic phased-array testing can easily control the acoustic beam and realize the deflection or focusing of the acoustic beam by applying different delays to each array element. It is widely used in the detection of welds and structurally irregular specimens [92]. In recent years, some scholars [93,94] have developed a new nonlinear detection method based on phased array testing. Its basic idea is still based on the nonlinear performances of ultrasonics at the damage location. That is, the acoustic kinetic energy in the fundamental frequency band is transferred to the energy in the high-frequency frequency band. However, the acoustic energy loss caused by nonlinear defects in ultrasonic phased arrays in different focusing modes is different. For example, in the modes of parallel focusing (all array elements are triggered) and group focusing (a single array element is triggered in turn, odd and even array elements are triggered separately, or the trigger unit is set according to the demand), the acoustic energy loss of the acoustic wave in the fundamental frequency band after nonlinear defects is different. Figure 20 is the schematic diagram of phased-array nonlinear imaging detection [95].

The system in Figure 20 is mainly composed of a transducer array, an ultrasonic front-end, and a display control unit. The system can support 32/64/128 array element transducer arrays. The ultrasonic front-end consists of a transmitting circuit, a receiving circuit, a receiving/transmitting isolation circuit, a high-voltage power supply, a high-voltage switch array, etc. The transmission circuit of the system can support the simultaneous transmission of up to 48 array elements. The transmit beamformer is realized by an FPGA (Field Programmable Gate Array), which generates the transmit pulse control signal according to the transmit focus parameters set by the host, including the transmit frequency, pulse number, pulse polarity, transmit aperture, and delay of various array elements, and sends it to the high-voltage pulse generation circuit. The receiving circuit includes a 48-channel low-noise amplifier, a variable-gain amplifier, a filter, a D/A converter, and a receiving beamformer; digital signal processing functions such as digital filtering, detection, and decimation filtering for beam output; and various timing controls. In the ultrasonic phased array nonlinear imaging detection mode, the central frequency and bandwidth of the bandpass filter and detector are matched with the nonlinear components (second harmonic components) in the received signal.

Gao et al. [95] developed a nonlinear ultrasonic phased array nondestructive testing system. Carbon steel specimens, a medical ultrasonic phantom, and a tungsten wire target were used as experimental materials. The testing results of the conventional linear ultrasonic phased array imaging method, the filter-based nonlinear phased array imaging method, and the inverse pulse nonlinear ultrasonic phased array imaging method are compared, and the comprehensive analysis shows that nonlinear ultrasonic phased array testing technology has better defect resolution and spatial resolution than traditional linear ultrasonic phased array testing technology. The spatial resolution of nonlinear phased array ultrasonic testing based on inverse pulse is significantly improved over that of nonlinear ultrasonic testing based on filter. Oharay et al. [96] used a commercial phased-array system and phased-array transducer to detect fatigue cracks in metal materials, collected experimental data on metal materials under different loading conditions, and extracted the second harmonic component for nonlinear imaging. The research results demonstrate that the imaging difference under different loads can characterize the specific situation of a fatigue crack in the specimen. Based on ultrasonic phased array technology, Potter et al. [97] realized the detection of an early fatigue crack in an A1 2014 plate by using the energy loss difference in the excitation bandwidth under parallel and serial focusing modes. The research indicates that this technology can effectively identify the early fatigue damage and is not affected by macro-sized defects (such as open cracks, holes, etc.); Haupert et al. [98] used the sum of the echo amplitudes under the two grouping trigger modes of phased array odd and even to subtract the echo amplitudes under the parallel focus mode to realize the detection of the fatigue closed crack defect in stainless steel AISI304. Jiao et al. [99] used the nonlinear imaging method based on the energy difference of the diffusion sound field under different phased array focusing modes to carry out ultrasonic inspection on the early fatigue damage of steel plates. The research results demonstrate that the use of fundamental wave energy loss to characterize various nonlinear effects caused by fatigue cracks can overcome the problem that the existing nonlinear Lamb wave detection methods are limited to a single nonlinear effect. Moreover, the difference in nonlinear response at the focal point of the two phased array focusing modes can effectively reveal the location of fatigue cracks.

At present, the ultrasonic phased array nonlinear imaging detection technology is relatively complex for signal processing [95,96,97,98,99,100], and real-time imaging is not possible. It needs to extract the signal and undergo a series of post-processing steps before positioning and imaging. The detection efficiency and accuracy need to be improved.

## 4. Key Technologies of Ultrasonic Nonlinear Testing

### 4.1. Sources of Nonlinearity

The key to nonlinear ultrasonic testing is to distinguish the nonlinearity caused by defects from the inherent nonlinearity of materials. In addition, ultrasonic transducers, power amplifiers, coupling agents, etc. are nonlinear, and these nonlinear effects are greater than the nonlinearity of defects and materials themselves. Therefore, it is very important to reduce unnecessary nonlinear effects in the test device and test method.

#### 4.1.1. Transducer

A pulse train should be used to excite the narrow-band piezoelectric chip transmitting transducer so that the excitation signal with a narrow frequency band can be obtained to extract useful information from the test results. Piezoelectric wafers can eliminate other nonlinear sources such as damping, protective covers, etc. In order to filter the front-end nonlinearity, it is necessary to measure the frequency response of the transmitting transducer and make full use of its excitation level and “filter” effect.

#### 4.1.2. Coupling Agent

Different liquid media are used as coupling agents for measurement, and the one with the smallest nonlinear residual is selected as the coupling agent, and its nonlinearity is quantitatively calibrated and corrected in the test results. Use a tightly coupled, low melting point, easy crystallization solid coupling agent or an adhesive to bond the wafer to reduce the nonlinearity of the coupling agent. The non-contact ultrasonic testing technology is used to overcome the nonlinearity introduced by the coupling agent and strengthen the research of air-coupled ultrasound, laser ultrasound, and electromagnetic ultrasound, but the attenuation of air to ultrasound should be considered.

### 4.2. Signal Processing and Calibration of Detection Results

#### 4.2.1. Signal Processing

For the higher harmonic technology, the amplitude of the second harmonic is very small, which is often difficult to distinguish from the noise signal. Therefore, it is necessary to strengthen the theoretical research in signal processing. The received signal is processed by a modulation method for bispectrum analysis or phase-sensitive detection to reduce the impact of noise on the detection effect. Using a broadband transducer with good frequency response as the receiving transducer, one can obtain the output signal with a large gain, and the fundamental wave, the second harmonic, and even the third harmonic have good dynamic response. In addition, to process the nonlinearity introduced by the preamplifier and the receiving transducer, the frequency selection function of the Ritec-SNAP system can also be used to amplify the specific frequency signal.

#### 4.2.2. Calibration of Test Results

According to Formula (9), the nonlinear coefficient *β* is related to the ratio of the second harmonic amplitude to the square of the fundamental wave amplitude, but the voltage data received from the transducer response is measured in the test. Therefore, it is necessary to calibrate the test data, that is, convert the voltage into displacement. The common calibration method is to directly measure the vibration displacement of the specimen using a laser vibrometer. Without calibration, meaningful test results can only be obtained after comparing the test results with the reference specimens. Due to the different frequency response characteristics of the receiving transducer, in order to make the measured data comparable, the voltage amplitude of the fundamental and higher harmonics should be compensated according to the frequency response transfer function. It is necessary to reasonably design the test device and plan the test steps, conduct signal processing, and analyze the test results to ensure the reliability of the test results.

## 5. Summary and Prospects

Nonlinear ultrasonic technology is an effective supplement and extension of traditional linear ultrasonic testing technology and has a good application prospect in the early damage detection of materials and mechanical property degradation [101,102,103]. Many studies and practices have proven that this technology has obvious advantages. At present, although some achievements have been made in this field, there is still room for improvement and development. List several aspects that need to be developed and improved in the next step.

(1) Nonlinear ultrasonic theory can be divided into two categories: classical nonlinear and contact nonlinear. The classical nonlinear ultrasonic theory starts with the discrete model and establishes the nonlinear ultrasonic wave equation in solid media. By solving the wave equation, the expression of the second-order nonlinear coefficient *β* is shown in Equation (9).

The classical nonlinear ultrasonic theory clarifies the influence mechanism of dislocation monopoles and dislocation dipoles and precipitates in the material microstructure on the ultrasonic nonlinear effect, and the total nonlinear coefficient caused by lattice strain, dislocation monopole displacement, and dislocation dipole displacement is shown in Equation (18).

The influence of the precipitation-dislocation pinning model on the nonlinear coefficient can be expressed as Equation (21).

The contact nonlinear theory mainly focuses on the ultrasonic nonlinear response on the solid interface, and the change of the spacing between two rough contact interfaces is related to the dynamic force of the interface and the acting force of ultrasonic. The tension/compression effect produced by the two interfaces of elastic rough contact is the main source of nonlinearity.

In the research of nonlinear ultrasonic theory, neither the early dislocation monopole model, the dislocation dipole model, nor the crack and interface state transformation spring model in recent years can independently characterize the whole fatigue process. Therefore, the application of nonlinear ultrasonic technology in metal fatigue life prediction is very limited. Strengthen the basic research of nonlinear ultrasonic testing technology and consider the influence of attenuation, noise, and other factors on the nonlinear response of the system; establish a more practical nonlinear model; expand the application range of nonlinear ultrasonic testing technology; and make it realize a leap from qualitative to quantitative, from simple characterization to life prediction.

(2) The frequency-mixing theory shows that the frequency ratio *α* and the included angle of the fundamental wave *Φ* can directly affect the propagation direction of the frequency-mixing wave *γ*. By adjusting the frequency ratio, the propagation direction of the frequency-mixing wave can be controlled. So far, the theoretical research on the mixing effects of body wave mixing (longitudinal wave and longitudinal wave mixing, longitudinal wave and shear wave mixing, shear wave and shear wave mixing, etc.) and Lamb wave mixing is relatively mature, and whether the mixing of other types of waves (such as surface wave, Love wave, interface wave, etc.) can produce mixing waves needs further research.

(3) It is the future development direction to combine nonlinear ultrasonic testing technology with big data and cloud computing to realize full-life and all-time structural health monitoring of important components. Based on a big data and cloud computing management platform, online monitoring of early damage to important components is realized. Through the fusion analysis and data mining of historical data, the remaining life of components and materials is predicted, thus helping us improve the safety and reliability of important components in service.

(4) Enhance the robustness of nonlinear ultrasonic testing technology. The application of this technology in engineering is limited due to the non-defect nonlinearity of temperature, impact, mechanical vibration, and other noises in the testing process. Therefore, in addition to improving the test methods and eliminating the influence of non-defect interference on the test results, how to enhance the nonlinear response of defects should be considered. Using the modulation method to capture beam aliasing is a good research idea.

(5) Carry out research on the signal processing and imaging of nonlinear ultrasonic testing technology. Dynamic photoelastic technology has been applied to the study of seismic waves, which can observe the propagation process of seismic waves in real time. Combining FPGA technology with dynamic photoelastic imaging technology, a high-precision multi-channel real-time imaging system is developed by embedding soft core technology, phase-locked loop technology, differential counting technology, serial communication technology, adjustable pulse width technology, etc., so as to realize the ultrasonic nonlinear imaging detection of early damage of metal materials, which is of positive significance to improving the efficiency and accuracy of non-ultrasonic imaging detection. It provides a new idea for the development of nonlinear ultrasonic imaging detection technology.

(6) In combination with practical engineering applications, optimize nonlinear nondestructive testing methods and develop testing equipment and methods suitable for complex working conditions in practical engineering applications.

## Figures and Tables

**Figure 1 materials-16-02161-f001:**
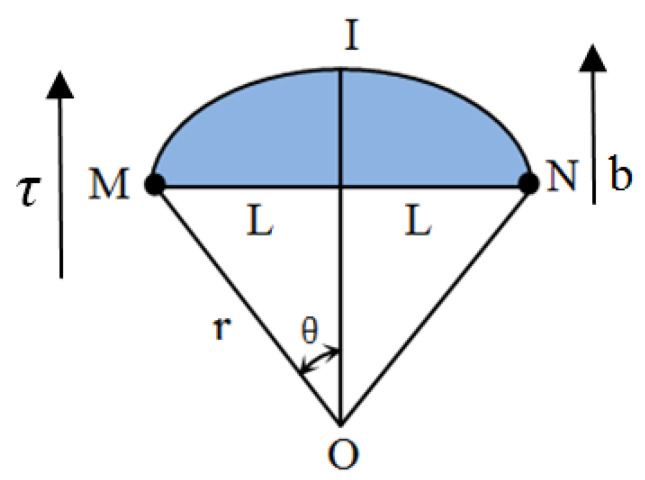
Dislocation monopole model.

**Figure 2 materials-16-02161-f002:**
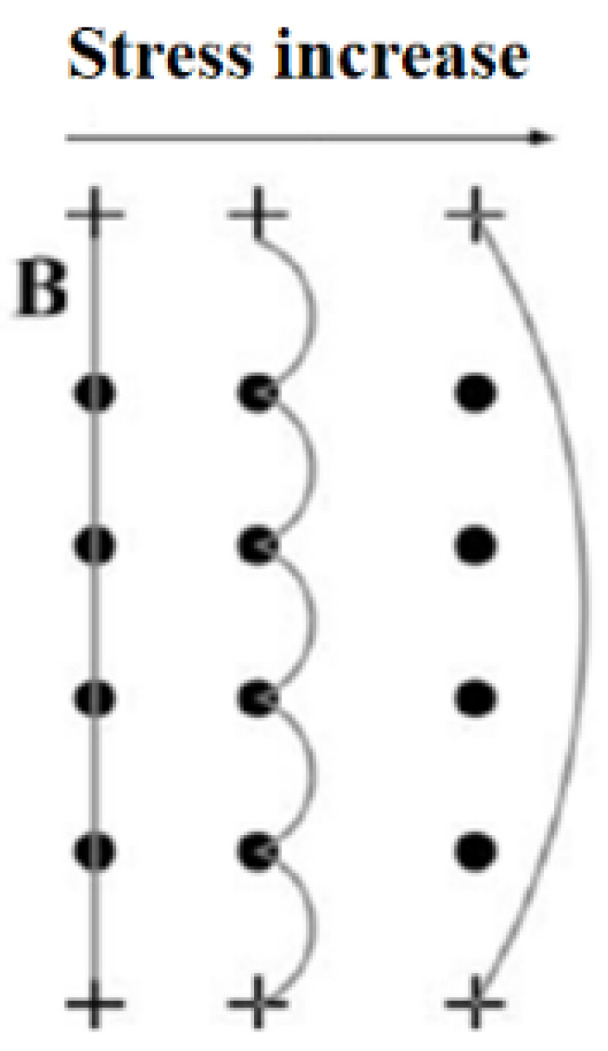
Dislocation movement.

**Figure 3 materials-16-02161-f003:**
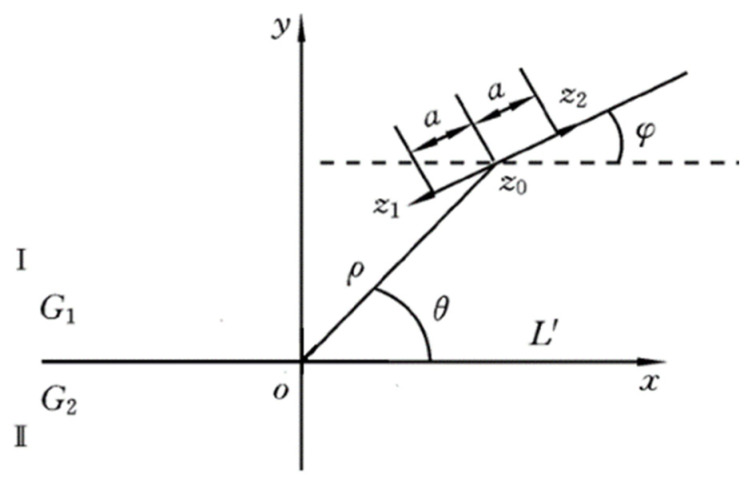
Dislocation dipole model.

**Figure 4 materials-16-02161-f004:**
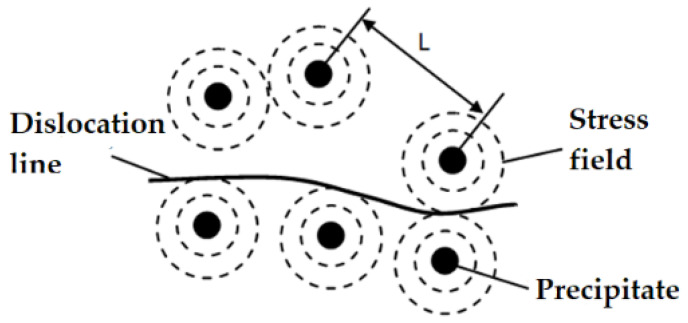
Precipitate-dislocation pinning model.

**Figure 5 materials-16-02161-f005:**
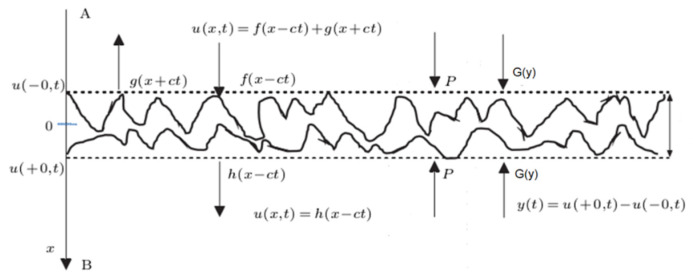
Schematic diagram of nonlinear ultrasonic response of microcracks.

**Figure 6 materials-16-02161-f006:**
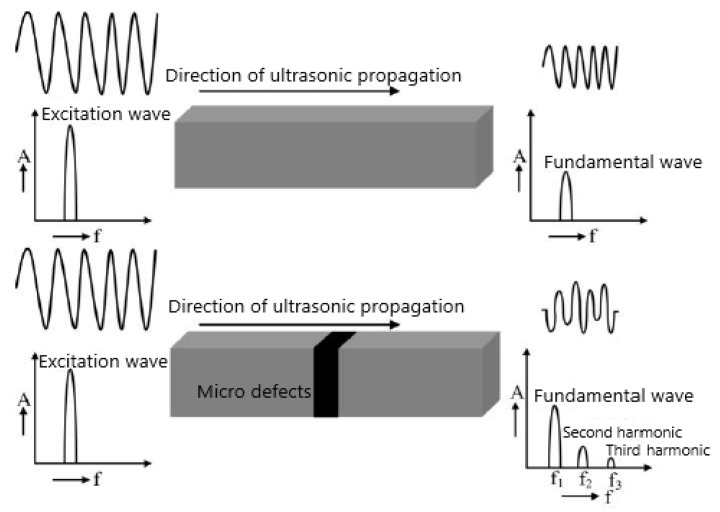
The principle of the higher-harmonic testing method.

**Figure 7 materials-16-02161-f007:**
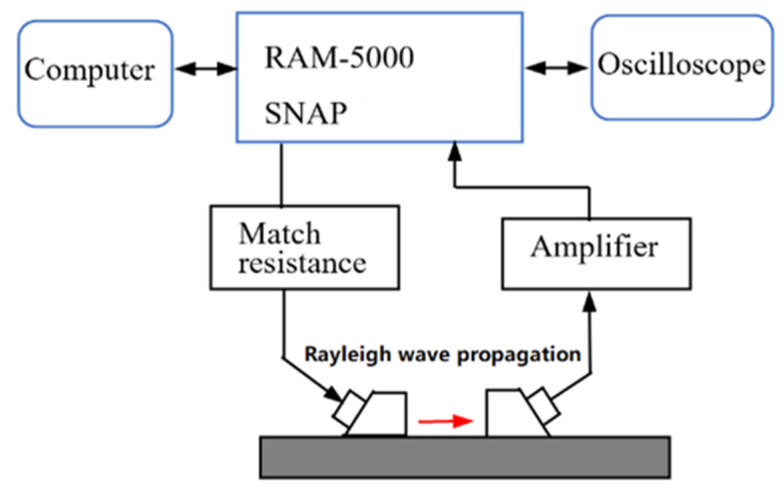
The nonlinear surface wave measurement system.

**Figure 8 materials-16-02161-f008:**
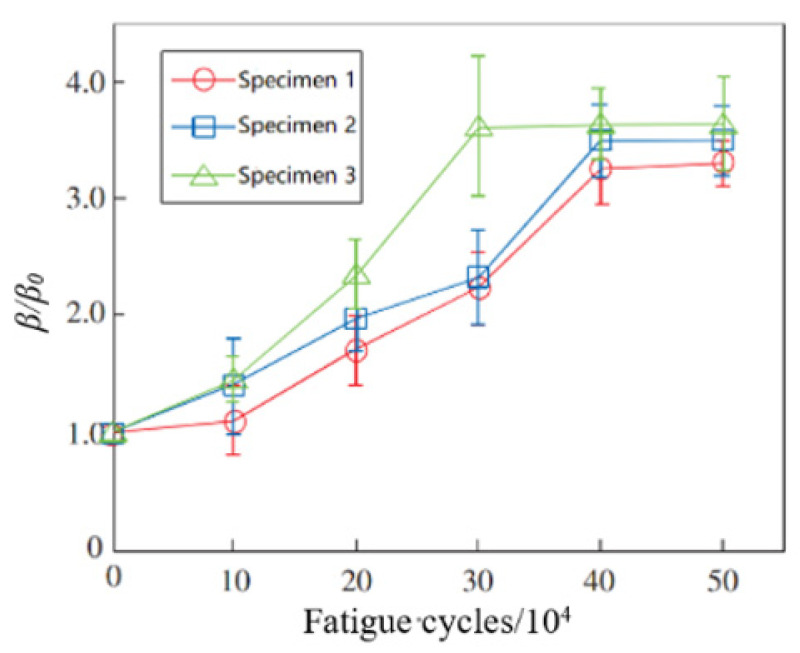
The relation curve between *β/β*_0_ and fatigue cycles.

**Figure 9 materials-16-02161-f009:**
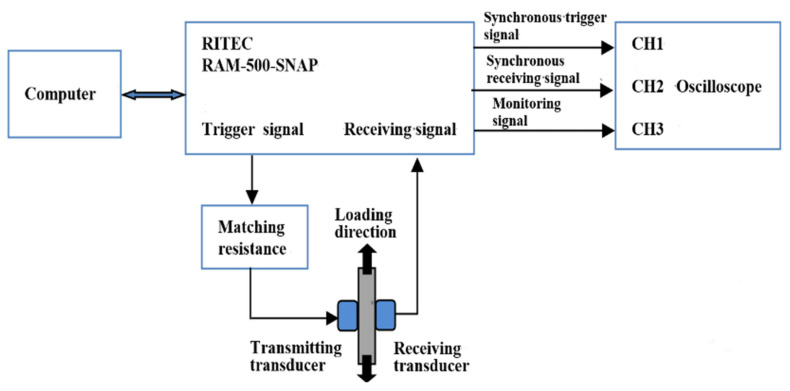
The nonlinear ultrasonic testing system.

**Figure 10 materials-16-02161-f010:**
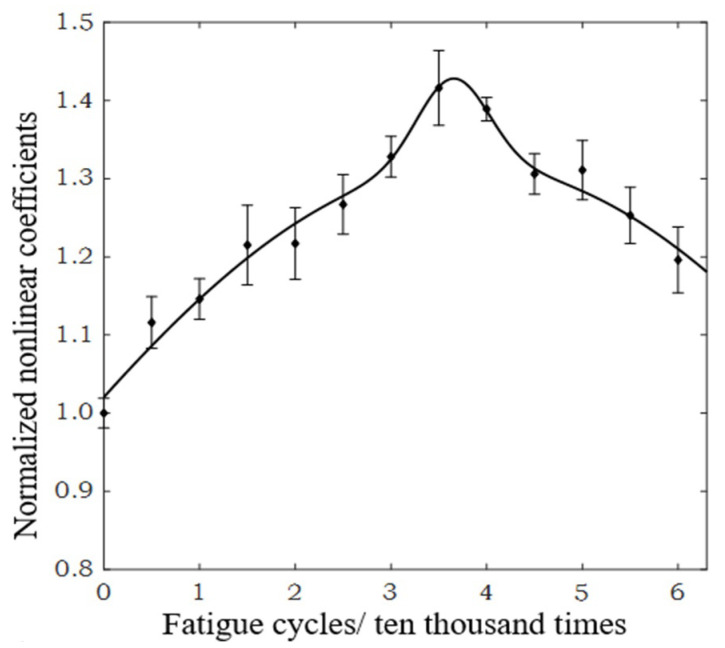
The relation curve between the normalized nonlinear coefficient and fatigue cycles.

**Figure 11 materials-16-02161-f011:**
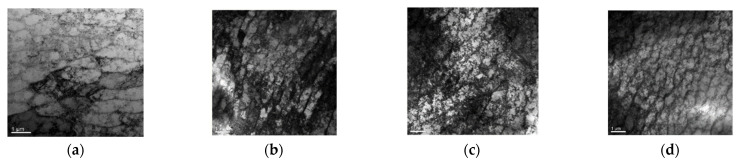
TEM photos of fatigue specimen under different cycles [51]. (**a**) Dislocation line (10,000 times); (**b**) Dislocation tangle (20,000 times); (**c**) Dislocation vein (30,000 times); (**d**) Dislocation wall (40,000 times).

**Figure 12 materials-16-02161-f012:**
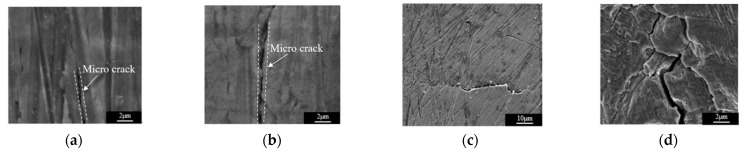
SEM photos of fatigue specimen under different cycles [51]. (**a**) 30,000 times; (**b**) 40,000 times; (**c**) 50,000 times; (**d**) 68,000 times.

**Figure 13 materials-16-02161-f013:**
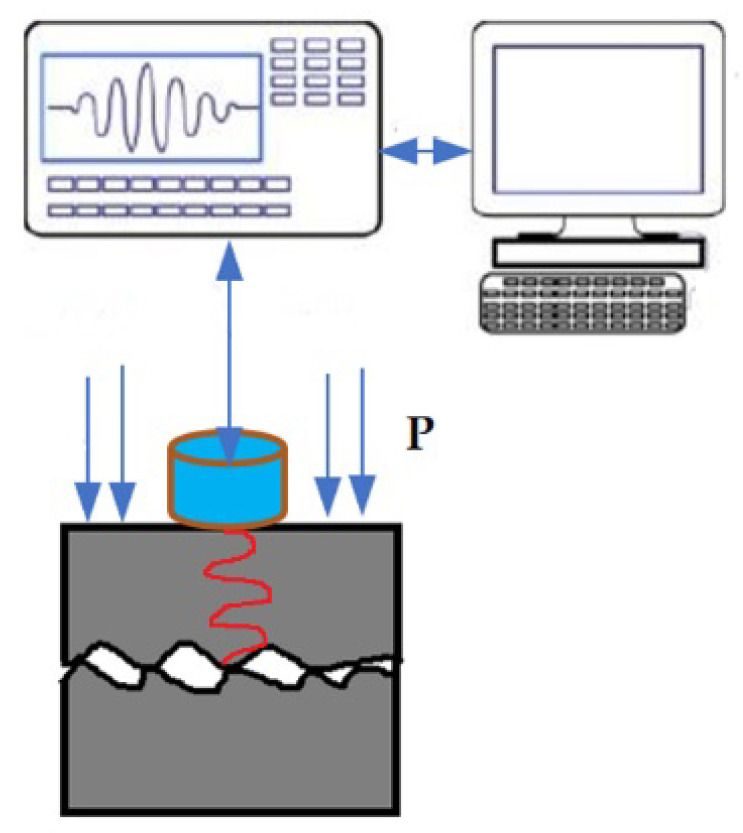
Interface nonlinear response model.

**Figure 14 materials-16-02161-f014:**
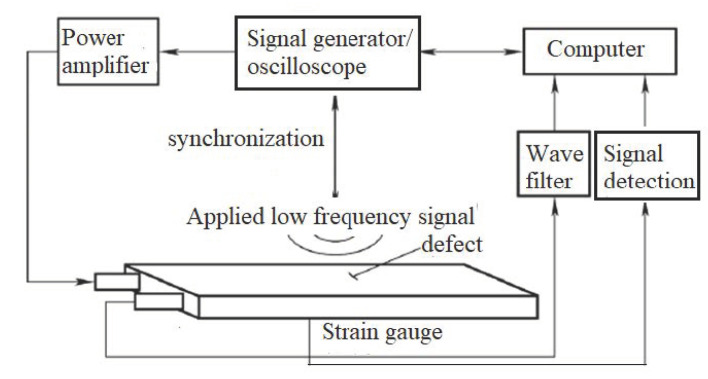
The principle of the vibration modulation technology.

**Figure 15 materials-16-02161-f015:**
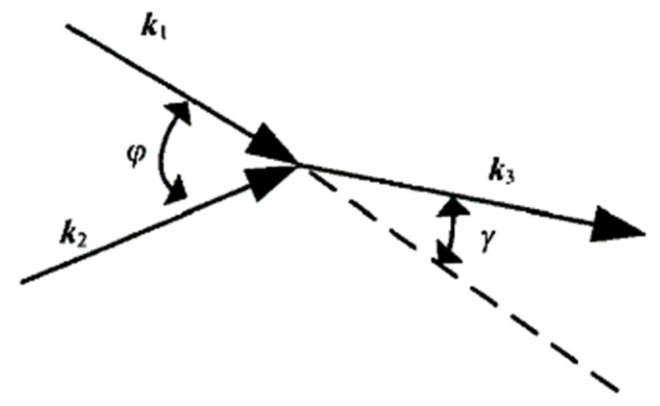
Diagram of interaction between two waves.

**Figure 16 materials-16-02161-f016:**
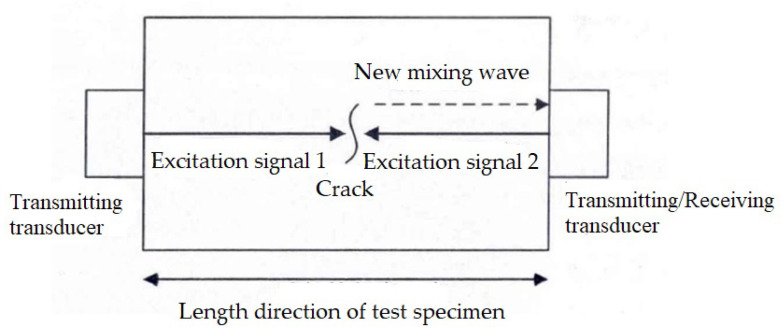
The principle of collinear wave mixing detection.

**Figure 17 materials-16-02161-f017:**
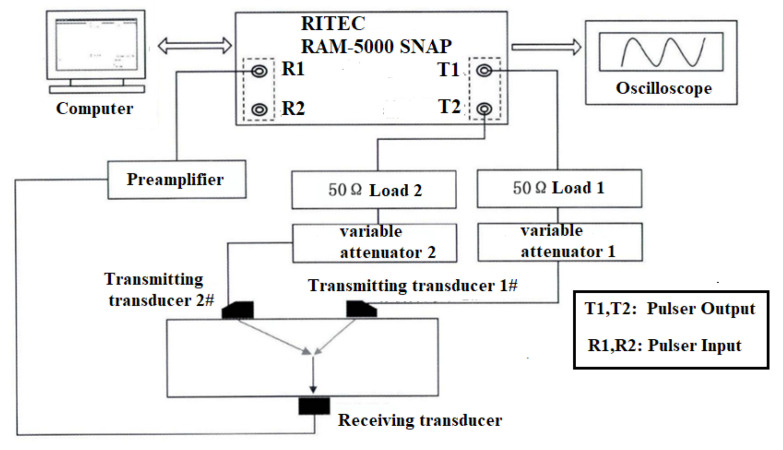
Schematic diagram of non-collinear wave mixing test device.

**Figure 18 materials-16-02161-f018:**
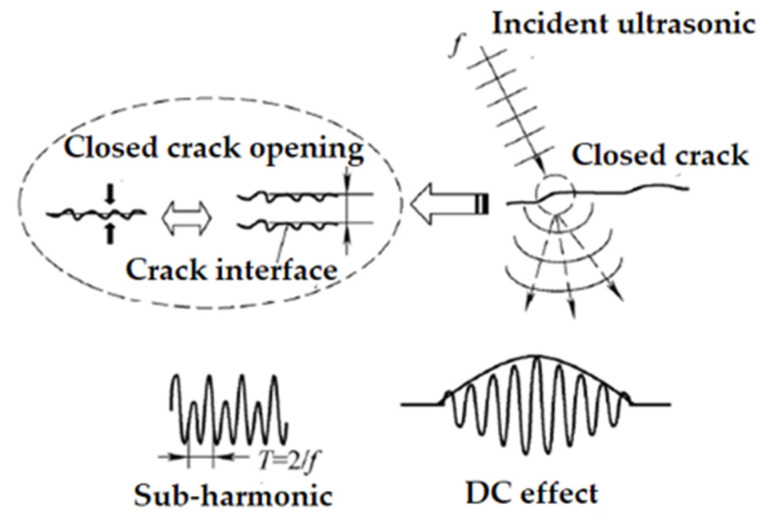
Generation of sub-harmonic and DC effects.

**Figure 19 materials-16-02161-f019:**
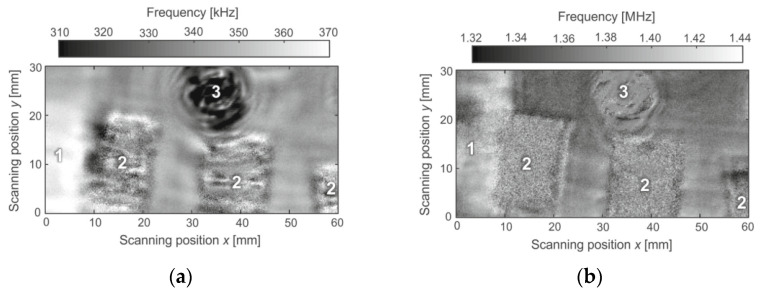
F-scan imaging results of defects in CFRP plate [91]. (**a**) F-scan of 0.31–0.37 MHz; (**b**) F-scan of 1.32–1.44 MHz.

**Figure 20 materials-16-02161-f020:**
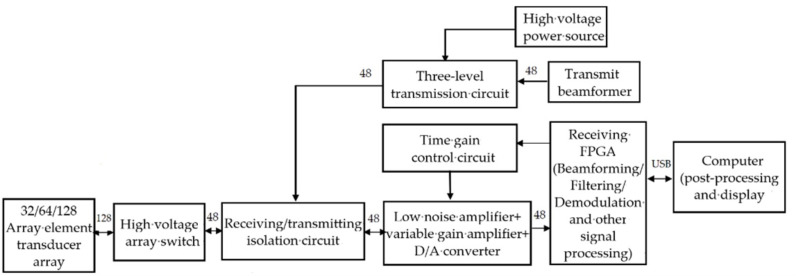
Schematic diagram of phased-array nonlinear imaging detection.

**Table 1 materials-16-02161-t001:** Conditions of mixing two waves to generate frequency-mixing wave.

Conditions	Incident Wave	Frequency-Mixing Wave	Direction of Frequency-Mixing Wave	cos*ψ*	Tan *γ*	Range of Frequency Ratio
1	*T*(*ω*_1_)*,T*(*ω*_2_)	*L*(*ω*_1_ + *ω*_2_)	*k*_1_ + *k*_2_	$c^{2}-\frac{\left(1-c^{2}\right)\left(1+a^{2}\right)}{2a}$	$\frac{a\sin\varphi }{1+a\cos\varphi }$	$\frac{1-c}{1+c}<a<\frac{1+c}{1-c}$
2	*L*(*ω*_1_)*,L*(*ω*_2_)	*T*(*ω*_1_ *− ω*_2_)	$\;\;\frac{k_{1}-k_{2}}{\omega_{1}-\omega_{2}}$	$\frac{1}{c^{2}}-\frac{\left(1-c^{2}\right)\left(1+a^{2}\right)}{2ac^{2}}$	$\frac{-a\sin\varphi }{1-a\cos\varphi }$	$\frac{1-c}{1+c}<a<\frac{1+c}{1-c}$
3	*L*(*ω*_1_)*,T*(*ω*_2_)	*L*(*ω*_1_ + *ω*_2_)	*k*_1_ + *k*_2_	$c-\frac{a\left(1-c^{2}\right)}{2c}$	$\frac{a\sin\varphi }{c+a\cos\varphi }$	$0<a<\frac{2c}{1-c}$
4	*L*(*ω*_1_)*,T*(*ω*_2_)	*L*(*ω*_1_ *− ω*_2_)	$\frac{k_{1}-k_{2}}{\omega_{1}-\omega_{2}}$	$c+\frac{a\left(1-c^{2}\right)}{2c}$	$\frac{-a\sin\varphi }{c-a\cos\varphi }$	$0<a<\frac{2c}{1+c}$
5	*L*(*ω*_1_)*,T*(*ω*_2_)	*T*(*ω*_1_ *− ω*_2_)	$\frac{k_{1}-k_{2}}{\omega_{1}-\omega_{2}}$	$\frac{1}{c}-\frac{\left(1-c^{2}\right)}{2ac}$	$\frac{-a\sin\varphi }{c-a\cos\varphi }$	$\frac{1-c}{2}<a<\frac{1+c}{2}$

## Data Availability

The data presented in this study are available on request from the corresponding author.

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
