# Peer review of "Research Progress in Nonlinear Ultrasonic Testing for Early Damage in Metal Materials"

_materials, 2023, doi:10.3390/ma16062161_

Round 1
Reviewer 1 Report
This manuscript studied the issue of “Research Progress of Nonlinear Ultrasonic Testing for Early Damage in Metal Materials”.
First of all, I would like to thank the authors of this manuscript for the effort they put into making it. The paper needs to be rewritten and its objectives well redefined. On the other hand, I have added some comments with the main objective of improving the manuscript. In my opinion, the subject of the paper is remarkably interesting. However, the paper needs some major revisions.
i. What is the innovation point or significance of the study for this article? Please make clear the novelty and contribution of the manuscript and its results as compared to the extensive literature available. The paper does not provide a clear objective of the study.
ii. . What is the difference between this manuscript and the article https://aip.scitation.org/doi/10.1063/1.4879415 and 10.3901/JME.2016.06.022 that was published before?
iii. Some latest relative papers are not included. Such as:
doi.org/10.1016/j.measurement.2021.110155
doi.org/10.12989/sss.2021.27.3.479
doi.org/10.1016/j.jmapro.2022.10.067
iv. The article form is a review paper form and specific improvements should be mentioned by the authors consider regarding the methodology
v. DC effect should be describing in text.
vi Results are merely present and there are no scientific findings are discussed. The introduction on similar work is very limited and does not cover similar experiences on the topic. I strongly recommend authors give a broader overview of similar works on the topic. The introduction should focus on the content related to the topic of the article.
vii. All the parameters used in the text should be defined. I suggest the authors provide a nomenclature.
viii. In conclusions, the useful data is not provided and the words prove general and lack of academic contributions. Please provide more specific details in the conclusions. The current conclusion is rather generic. The discussion is rather basic and short.
ix. The English language could be improved as well as the format of the manuscript; though overall English is acceptable.
Author Response
Dear Reviewer:
Thank you very much for the time and effort that you have put into reviewing the previous version of the manuscript. Your suggestions have enabled me to improve my work greatly. Appended to this letter is my point-by-point response to the comments raised by you. The comments are reproduced and my responses are given directly afterward in a different color (red).
Should you have any questions, please contact me without hesitate.
Kind regards
Yan XiaoLing

Reviewer 2 Report
Dear authors, thank You for the interesting research. The subject of early fatigue damage detection plays a major role for the industry. Especially non destructive methods are important. Due to the fact that the authors are analysing non-destructive methods in the frequency domain it would also be wise to include fatigue damage estimation models based on the Dirlik's, Benasciutti's or Böhm's papers which take into account the power spectral density in order to calculate the damage till the crack initiation occurs. Nevertheless the paper is interesting and will serve to a wide audience of both engineering and scientific readers. The literature review is solid and up to date. The citation style is not the correct one for materials, please follow the template. The methods are well described, the text needs a lot of revision as for example the small letters after the numbers describing next parts of the paper.
The whole text needs to be revised as there are a lot of minor mistakes like e.g:
page 3: "Dislocation is linear defect in the material structure"
page 4: "It can be seen from formula (16).."
page 6: "Figure 5 is the schematic diagram of the microcrack model"
page 11: "When nonlinear frequency-mixing technology is applied, if two incident waves are parallel to each other or on the same axis, this technology is called collinear wave mixing technology."
etc.
Please give the text to be double checked as in terms of english.
I'm missing the classical discussion and results part as they are streched across the last parts of the paper. The presented conclusions are clear and understandable.
I suggest major revision.
Author Response

(The authors gave the same response as above.)

Reviewer 3 Report
The Authors presented in a condensed version the development of non-linear tests using ultrasonic methods used to assess defects in metal materials. The content of the manuscript is properly edited. The main methods of this technique used to detect all kinds of defects resulting from the so-called fracture mechanics. The publication of this manuscript is absolutely justified because it is structured knowledge that can be helpful to other researchers in continuing to develop these methods, but also to researchers focusing on the propagation of fatigue cracks, etc. Before the manuscript is published, the reviewer recommends making minor editorial corrections without re-reviewing. Check the equations, especially in the paragraph after equation (14). Especially to emphasize the purpose of this article because in the section Introduction it is blurred.
Author Response

(The authors gave the same response as above.)

Reviewer 4 Report
The following modifications are suggested before publishing the manuscript.
Introduction
Page 1 : ‘... it is very important to detect the early damage of metal parts before the crack occurs.’ The problem is often the resolution of the detection method. Studies in the 1980’s have demonstrated that crack initiation can be defined with crack size of about 10 micrometers. (using Replica method for example: Ritchie, R O, and Lankford, J. Small fatigue cracks; Proceedings of the Second International Conference/Workshop, Santa Barbara, CA, Jan. 5-10, 1986). Then the crack propagation is well predicted by the fracture mechanics.
However replica method is not suitable to detect and follow cracks in industrial components. Thus the need for accurate and reliable methods such as nonlinear ultrasonic techniques.
Page 3 : ‘The pinning points can be grain boundaries, other dislocations or point defects in materials.’ If referring to the configuration shown in figure 1, the pinning points cannot be grain boundaries!
The mechanism the authors are referring was proposed by Frank, F. C.; Read Jr, W. T. (1950). "Multiplication Processes for Slow Moving Dislocations". Physical Review. 79 (4): 722–723. doi:10.1103/PhysRev.79.722
Page 5 : figure 3. Do you have a reference for the model of the dislocation dipole.
Is your definition of dislocation dipole the same as that proposed by James Li (
Li, J. C. M. (1964). Interaction of dislocation dipoles. Discussions of the Faraday Society, 38, 138. doi:10.1039/df9643800138)?
Page 5: ‘When r >ra, the stress around the lattice is σr.’ The hydrostatic stress?
Page 5 : Its size can be expressed as’. The authors might mean : the amplitude?
Page 8 : the authors wrote : ‘With the deepening of fatigue damage, the ultrasonic nonlinear coefficients increased to the extreme point and then began to decline. The reason is as shown in Figure 9 (b), (c) and (d). In the microstructure of the material, there are non-planar dislocation structures such as dislocation veins, dislocation walls, dislocation cells and so on, which are dominated by dipole distribution.’
However increase in the dislocation density and reorganization of the dislocation into dislocation cells are not damage. Such modifications of the microstructure are reversible. Damage by crack formation is not reversible.
According to the authors, can the increase and decrease in the signal shown in Figure 9 be interpreted by, respectively, increase in the dislocation density and damage ? This should be mentioned.
English:
Introduction:
Page 1 : ‘ ... the detection effect of early damage of metal materials is not good.’ Please rewrite the last part of the sentence!
Page 1 : ‘... the remaining fatigue life of parts may be less than 20%.’ remaining fatigue
Page 2 : ‘Simultaneous Equations (1)–(3) , ignore the higher-order terms (higher than the second-order) in the equation, and the resultant nonlinear wave equation can be ex-pressed as:’ This sentence is not clear. Please rewrite it.
Page 3 : ‘Dislocation is linear defect...’ Articles are missing : A dislocation is a linear defect... Better write in plural form: Dislocations are linear defects...
Page 9 : ‘To sum up...’ : Please write in proper English!
Page 10 : 3.3. Frequency (capital F)
Page 11 : 3.3.1. Collinear (capital C)
Page 15 : 3.5. Ultrasonic (capital U)
Page 18 Summary and Prospective(s)
Author Response
Dear reviewer:
Thank you very much for the time and effort that you have put into reviewing the previous version of the manuscript. Your suggestions have enabled me to improve my work greatly. Based on your comment and request, I have made extensive modification on the original manuscript. Here, I attached revised manuscript in the formats of PDF, for your approval. Appended to this letter is my point-by-point response to the comments raised by you. The comments are reproduced and my responses are given directly afterward in a different color (red).
A revised manuscript with the correction sections red marked was
attached as the supplemental material and for easy check/editing purpose.
Should you have any questions, please contact me without hesitate.
best regards
Yan XiaoLing

Round 2
Reviewer 1 Report
The paper can be accepted for publication in its present form.
Reviewer 2 Report
Dear authors, thank You for the revised version. I suggest to accept the paper.